# Choosing the Best Tissue and Technique to Detect Mosaicism in Fibrous Dysplasia/McCune–Albright Syndrome (FD/MAS)

**DOI:** 10.3390/genes15010120

**Published:** 2024-01-18

**Authors:** Yerai Vado, Africa Manero-Azua, Arrate Pereda, Guiomar Perez de Nanclares

**Affiliations:** Rare Disease Research Group, Molecular (Epi) Genetics Laboratory, Bioaraba Health Research Institute, Araba University Hospital, 01009 Vitoria-Gasteiz, Spain; yerai.vado@bioaraba.org (Y.V.); africa.manero@bioaraba.org (A.M.-A.); arrate.peredaaguirre@osakidetza.eus (A.P.)

**Keywords:** *GNAS*, genetic mosaicism, Fibrous Dysplasia/McCune–Albright syndrome, Sanger sequencing, Next-Generation Sequencing

## Abstract

*GNAS*-activating somatic mutations give rise to Fibrous Dysplasia/McCune–Albright syndrome (FD/MAS). The low specificity of extra-skeletal signs of MAS and the mosaic status of the mutations generate some difficulties for a proper diagnosis. We studied the clinical and molecular statuses of 40 patients referred with a clinical suspicion of FD/MAS to provide some clues. *GNAS* was sequenced using both Sanger and Next-Generation Sequencing (NGS). We were able to identify the pathogenic variants in 25% of the patients. Most of them were identified in the affected tissue, but not in blood. Additionally, NGS demonstrated the ability to detect more patients with mosaicism (8/34) than Sanger sequencing (4/39). Even if in some cases, the clinical information was not complete, we confirmed that, as in previous works, when the patients were young children with a single manifestation, such as hyperpigmented skin macules or precocious puberty, the molecular diagnosis was usually negative. In conclusion, as FD/MAS is caused by mosaic variants, it is essential to use sensitive techniques that allow for the detection of low percentages and to choose the right tissue to study. When not possible, and due to the low positive genetic rate, patients with FD/MAS should only be genetically tested when the clinical diagnosis is really uncertain.

## 1. Introduction

*GNAS* is a gene that encodes for the α subunit of the stimulatory G protein (Gsα). Gsα is part of a heterotrimeric G protein that binds to G-protein-coupled receptors (GPCRs). When the hormone binds to its GPCR, this causes a conformational change in the GPCR, which allows it to act as a guanine nucleotide exchange factor (GEF). The GPCR can then activate the Gsα by exchanging the guanosine diphosphate (GDP) bound to the G protein for a guanosine triphosphate (GTP). The activated Gsα can then dissociate from the β and γ subunits. Following this, the Gsα activates adenylyl cyclase, which stimulates the production of cyclic AMP (cAMP). cAMP is a second messenger, which finally activates cAMP-dependent protein kinase A (PKA), which phosphorylates different target proteins [1,2] (Figure 1).

Researchers have identified activating and inactivating mutations, either germinal or somatic, in the *GNAS* gene that impair its function. The vast majority of the variants found in patients are inactivating, with more than 400 unique variants reported [3], and they are associated with pseudohypoparathyroidism type 1A (PHP1A), pseudopseudohypoparathyroidism (PPHP), or progressive osseous heteroplasia (POH) (for review [4]), which is currently renamed as inactivating PTH/PTHrP signaling disorder type 2 (iPPSD2) [2]. A dual effect has been described for a missense variant (p.A366S) affecting both the stability and the activity of the Gsα in a patient with typical PHP1A features combined with testotoxicosis [5,6].

On the other hand, *GNAS* gain-of-function mutations lead to the constitutive production of cAMP, independent of the ligand binding to the receptor, and, consequently, dysregulated downstream signaling [7,8,9]. In the germline, they have recently been associated with Nephrogenic Syndrome of Inappropriate Antidiuresis (NSIAD), where patients present an impaired capacity to release water in urine, hyponatremia, skeletal developmental problems, and precocious puberty [10,11]. 

Endocrine or non-endocrine tumors have been identified when the activating variants are somatic, in which the *GNAS* gene acts as an oncogene [7]. Moreover, they can also cause a spectrum of diseases, known as Fibrous Dysplasia/McCune–Albright syndrome (FD/MAS; OMIM#174800), which is a rare disorder. Although OMIM#174800 refers to McCune–Albright syndrome as the main term, it includes FD among the other entities represented in the entry. Additionally, since 2017 [12], the international consortium for the study of these disease (https://www.icfdmas.com/, accessed on 8 January 2024) has referred to it as FD/MAS and promotes its use. Its exact prevalence is unknown, but it is estimated that FD is responsible for approximately 5% of benign bone lesions [13]. The first visible features are cutaneous affections, specifically *café au lait* spots on the skin with jagged and irregular borders. Secondly, the bones are also affected, shown as fibrous dysplasia of the bone (FD); in monostotic fibrous dysplasia, only one skeletal site is affected, and when more than one site is involved, FD is classified as polyostotic [8]. The last distinctive sign is peripheral precocious puberty (PP), which is less common in boys than in girls [8,14,15,16,17]. Other manifested signs include hyperthyroidism, growth hormone (GH) excess, and Cushing’s syndrome [16]. Due to its heterogeneity and phenotypic variability, the clinical diagnosis and management of FD/MAS is very challenging. The current guidelines summarize all of the clinical and molecular characteristics of this disorder and indicate that a clinical diagnosis of MAS can be established by the presence of the combination of FD and one or more extra-skeletal features or by the presence of two or more extra-skeletal features [8].

Genetically, FD/MAS is mainly associated with variants occurring in the Arginine in position 201 (R201) of the Gsα protein [18], which can be substituted by two different amino acids: Cysteine (NM_000516:c.601C>T, p.R201C) or Histidine (NM_000516:c.602G>A, p.R201H) [19]. A mutation in the 227 codon replacing Glutamine with Lysine (NM_000516:c.680A>T, p.Q227L) [19] has been identified in approximately 5% of patients [20]. 

As FD/MAS occurs in a sporadic manner, and due to the pattern of the *café au lait* pigmentation of the skin, the scientific community suspected that FD/MAS follows a somatic dominant inheritance [21]. In fact, the activating mutation in *GNAS* occurs postzygotically, and the resulting individual is a genetic mosaic [22] (Figure 2).

Genetic mosaicism is the existence of two or more genetically different cell types within one single tissue or individual [23]. That is, an organism is created from one zygote that presents genotype heterogeneity [24,25]. To form a mosaic, a de novo mutation in the genome occurs during the first stages of development [26]. As the mutation is postzygotic, different cell lineages will develop, and that is why each tissue of the organism will have one different genotype [25] (Figure 2). Because in mosaic organisms, only some tissues are affected, the choice of the adequate tissue to proceed to a correct diagnosis is mandatory [8,23,27]. If an unaffected cell sort is taken, the mutation will remain hidden, and a wrong diagnosis could be given [28,29].

Because of the somatic mosaic nature of the disease, some variants can only be observed in very low levels. Due to this problem, conventional sequencing technologies, such as Sanger sequencing, cannot detect mosaics below 10% [27] or 15–20% according to other authors [30]. Errors in detecting mosaicism have direct consequences in clinical practice [30]. On one side, we can issue a normality diagnosis in a patient, rejecting the clinical suspicion [28,31,32]. On the other side, mosaicism in progenitors can be a cause of disease in the offspring. However, if mosaicism is not detected in the parents, there will be a wrong de novo diagnosis [28,33,34]. Therefore, the genetic counselling will be compromised.

To overcome the problem of the detection limit of Sanger sequencing, other techniques that are more sensitive and can detect mosaicism more accurately have been developed. One of the most used techniques is Next-Generation Sequencing (NGS), thanks to its deep coverage [27,35]. Another more unknown technique is droplet-digital polymerase chain reaction (ddPCR). In this case, specific variants are interrogated and can detect even very-low-percentage mosaics [27]. This has been used to identify mosaicism in several diseases [36,37,38], including in FD/MAS [19,39]. In other rare diseases, quantitative real-time PCR (qRT-PCR) has also been used for the detection of mosaicism [40].

Taking all of the above into account, the main goal of this study is to clinically and genetically characterize all patients with a clinical suspicion of FD/MAS received in the laboratory.

## 2. Materials and Methods

### 2.1. Sample Selection

We included all probands with FD/MAS suspicion that have been referred to the Molecular (epi)Genetics Laboratory of the Araba University Hospital—Bioaraba Health Research Institute (Vitoria-Gasteiz, Spain) for *GNAS* molecular analysis since 2010. Different tissues (blood, bone, skin, etc.) were analyzed according to the availability.

### 2.2. Nucleic Acid Extraction

Genomic DNA of the patients was extracted from peripheral blood or different tissues using the QIAamp DNA Mini Kit (QIAGEN, Hilden, Germany), following the corresponding manufacturer’s instructions. 

### 2.3. Molecular Analyses

The presence of point variants in GNAS was investigated using Sanger sequencing as previously described [4]. Next-Generation Sequencing was performed with a custom panel as previously described [41].

### 2.4. Bioinformatic Approach

As a first-tier approach, the MiniSeq integrated DNA Enrichment Analysis Module was used for secondary analysis (alignment with BWA 0.7.7 on GRCh37/hg19 and Variant Call Format (VCF), and bam/bai files were obtained with Somatic Variant Caller v3.5.2.3 [42] and SAMtools v0.1.19, respectively). Then, output bam and VCF files were visualized with Integrative Genomics Viewer (IGV) [43] to check for the presence of variants associated with FD/MAS. 

## 3. Results

A total of 40 patients were referred to our center, and their *GNAS* gene was sequenced to look for variants associated with FD/MAS. The main results of the available clinical data and the genetic studies are summarized in Table 1.

According to the received clinical information, 6 patients presented isolated fibrous dysplasia, whereas 17 satisfied the clinical criteria for MAS. The rest of the patients had some of the characteristics of MAS in isolation and, probably, a screening test was requested for the genetic study. It must be outlined that four females had ovarian cysts, two patients had hyperthyroidism, one had GH excess, and one patient had neonatal Cushing’s syndrome. 

Regarding the samples, only blood (or DNA extracted from blood) was received from 17 patients, only the affected tissue (either fresh (6) or paraffin-embedded (1)) was received from 7 patients, and from 16 patients, both blood and the affected tissue (either fresh (13) or paraffin-embedded (3)) samples were received.

Sanger sequencing did not detect genetic variants in the DNA extracted from peripheral blood, but in four affected tissues (two bones, one pituitary, and one thyroid), the causative variant was identified. Of these, two carried the p.R201C mutation, and the other two carried the p.R201H mutation.

Regarding the NGS results, in seven samples, the variant was called (i.e., present in the variant calling file, shown in dark grey in Table 1) using the integrated software. During the visualization of these variants by IGV, we noticed that the other samples (shown in light grey in Table 1) also showed a possible presence of genetic alteration with a frequency of ≤4%, which is below Sanger’s detection limit [27].

So, finally, NGS was able to identify the presence of the mutation in four of the blood samples analyzed from all cases in a very low mosaicism percentage (<5%). Regarding the tissues, in four of them, the p.R201C mutation was detected, and in the other three, the p.R201H mutation was detected. Of all of the positive tissues examined using NGS, three samples were also detected by Sanger sequencing, probably due to the high percentage in which the variant was present. In one of the samples identified as positive using Sanger sequencing, the NGS sequence quality was not good enough (<30× GS1189 in Table 1) to consider the identification of the variant as positive. This low depth was probably caused by the tissue being paraffin-embedded.

When correlating the identification or lack of identification of the variant using any of the sequencing techniques with the patient’s clinical data, we observed that it was identified in four of the six patients with FD, in five of the seventeen patients with MAS, and in a skin sample from a patient for whom no clinical data were sent when the genetic study was requested.

## 4. Discussion

Fibrous Dysplasia/McCune–Albright syndrome (FD/MAS) comprises a spectrum of disorders caused by activating somatic mutations in the *GNAS* gene [16]. As the genetic variants occur postzygotically at any stage of development, the patient may have the variant in some of its cells and not in others [23,26]. In general, in somatic mosaicism, different mosaic ratios may be found in a patient’s distinct body tissues. This means that, depending on the tissue analyzed and the sensitivity of the technique used, alterations that are actually present may not be identified (false negatives).

Currently, there are several techniques that allow for the detection of mosaicism [19,35,44,45]. In this research work, we tried to identify somatic activating pathogenic variants that cause FD/MAS in a cohort of patients received in the laboratory. Using conventional Sanger sequencing, the variant was identified in four tissue samples but in none of the blood samples. On the contrary, when the same samples were studied using NGS, the variants were detected in six affected tissues (including three of those identified through Sanger sequencing) and five blood samples.

This difference in the results obtained using the two techniques has been acutely reported in the literature [27,30]. Even though Sanger sequencing has been considered the gold-standard, it has been deeply demonstrated that its detection limit is compromised; mosaics below 10–20% are hardly detected and, in most cases, they may not be seen [27,30]. To overcome this outcome, in recent years, the tools and the sensitivity needed to detect mosaicism have increased substantially. To detect variants at the genetic level (short deletions/insertions and single-nucleotide variants (SNVs)), droplet-digital polymerase chain reaction (ddPCR) and real-time PCR using TaqMan probes [19,46] have been used. 

On the other side, NGS produces single sequence reads, which enhances sensitivity and allows for a quantification of mosaicism. As the single reads can be counted, a precise percentage of mosaicism can be calculated, even in very low percentages [27,35]. In the present work, the sensitivity of the technique allowed us to find the mutation in seven samples (six different patients) in which the mosaicism was observed in very low levels (≤5%), which means that there are very few reads of the altered allele. As the number of reads in these mosaics is low, it is usually difficult to differentiate between a sequencing artifact and a real mosaic. The somatic variant caller uses a Poisson model to establish the Q score of a variant, and it excludes the variant if it presents a Q score below Q20 (which means there is a 1/100 probability of obtaining a false positive) [42]. Based on this model, it is not possible to call a 1% variant allele frequency (VAF) variant with confidence if the coverage is 200×, as the error rate is 1% (2/200 expected miscalls), i.e., there is an important overlap between noise and 1% VAF true variants. Probably, a coverage of 500–600×would enable variants below 5% VAF to be called. Of course, in a case where there is access to different tissues from the same person and there is more than one tissue confirming the presence of the variant, the confidence regarding the presence of the variant is high. But, if not, one way to solve this issue is to analyze the number of reads in each sense; if the reads are approximately the same, this provides an idea that it is a real variant, and not an artifact [47]. It is also important to confirm that the identified variant has a quality score over Q20 and that the quality of the reads containing it is good [48]. Moreover, the fact that the mutation identified is the one specific to that pathology also provides confidence, as seen in all of our samples.

Various biological materials may be tested in mosaicism investigation. Primarily, low invasive procedures are recommended in the sample collection process. Blood is one of the most easily obtained samples, and the collection procedure is not very invasive. However, genetic alterations in a mosaic are not usually present (or found) in blood. In this study, the variants were found only in 4 out of the 33 blood samples, which means there was a 12.12% rate of success (if all of the patients really present FD/MAS). On the other hand, 23 affected tissues were studied and, in 8, the variant was identified, which translates into a 34.78% success rate. As described previously, blood is not a good tissue to find mosaicism, as it is very rare to find it in blood [27,30]. So, in order to obtain more accurate results, it is important to study the affected tissues, even if obtaining samples is not so easy. Some studies also point to the preference of using biopsies more than blood in FD/MAS [27,30]. Nevertheless, we have to take into account that determining the total extent of the affected tissue is not an easy task [8]. Furthermore, different tissues may show different degrees of mosaicism levels, and if the biopsy includes normal tissue, false negatives may occur [8,35], i.e., if the sample is not adequately selected, the results that are obtained from the genetic study may not correspond to reality. Furthermore, we must not forget that it is not only important to choose the right tissue to study, but also the quality in which it has been preserved. In this study, several of the tissues were embedded in paraffin, which generated massive sequences of poor quality (with a low reading depth), which prevented us from considering the findings of the variants as true positives [8,49].

However, we must keep in mind that maybe some of the negative results can be true negatives (due to misdiagnosis) instead of false negatives (due to technical problems). For example, the success rate in identifying the genetic variant was higher in the samples received from patients with suspected FD than those received with extra-skeletal features. In fact, in some of the samples we received, the *GNAS* test was requested as a screening test, as the patients were still very young. However, just in this type of patients, it is important to bear in mind that the fewer symptoms present, the more difficult it seems to be to identify the presence of the mutation [19], even if essential for a clinical diagnosis that is really difficult in children who only present a single manifestation, such as hyperpigmented skin macules or precocious puberty. In these patients, both the clinical and the molecular diagnoses are difficult to confirm.

In brief, despite the improvements in techniques or in the selection of tissues to be analyzed, our data and those of other teams confirm that the success rate in detecting mosaicism in patients with MAS depends on (i) the severity of the disease, (ii) the tissue(s) studied, and (iii) the technique used. So, as already proposed in the international guidelines, a genetic study is only recommended when the diagnosis is uncertain, especially in patients with monostotic fibrous dysplasia without other skeletal or extra-skeletal abnormalities [8]. 

With the arrival of new genetic and genomic technologies, it may be worthwhile to continue researching and validating their usefulness for the diagnosis of this disease in different tissues, as suggested in the novel proposal presented by Roszko et al. [19]. It may also be advisable to update the current clinical diagnostic guidelines (mainly in the early stages of the disease) for better diagnosis and follow-up for these patients.

## 5. Conclusions

Genetic mosaicism (especially in low frequency) is difficult to detect using conventional techniques, such as Sanger sequencing. Due to its capacity to detect mosaicism, NGS is a reliable tool that facilitates the detection of mosaicism, even in very low levels, and it is better when analyzing the affected tissue. Nevertheless, it is important to establish a good bioinformatic pipeline to detect these variants and to differentiate real variants from artefacts.

## Figures and Tables

**Figure 1 genes-15-00120-f001:**
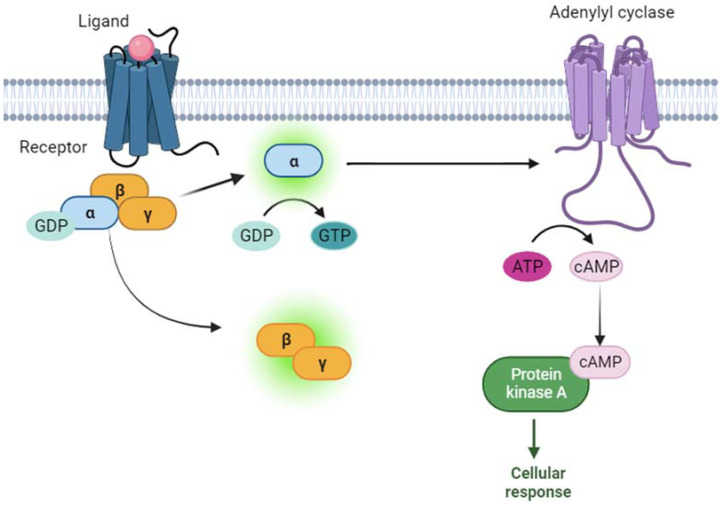
Signaling cascade of Gsα protein is schematically represented. Briefly, when the ligand (pink ball) binds to the receptor (blue), the α subunit of G protein (Gsα) exchanges guanosine diphosphate (GDP) with guanosine triphosphate (GTP). The activated α subunit activates adenylyl cyclase (purple), which produces cyclic adenosine monophosphate (cAMP) from adenosine triphosphate (ATP). Finally, cAMP activates protein kinase A (green), which activates different cellular effectors, generating a cellular response.

**Figure 2 genes-15-00120-f002:**
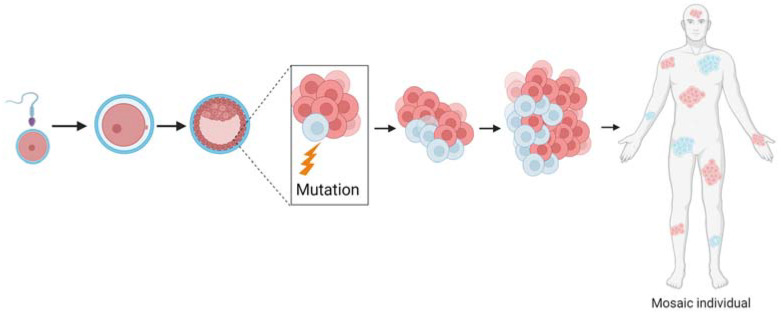
Genetic mosaicism is generated when a mutation occurs in early stages of development. As the genetic variant occurs postzygotically, there will be two or more cell populations. This means that, in the same individual, there can be at least two different genotypes.

**Table 1 genes-15-00120-t001:** Summary of the clinical and genetic results of patients with FD/MAS suspicion. -: no sample available; F: female; M: male; GH: growth hormone; ND: no data; *: tissue is paraffin-embedded. Samples with a positive genetic result are highlighted in grey (dark grey are samples detected with the integrated software, and light grey are samples detected with IGV).

Patient Code	Sex	Age at Study	Skeletal Abnormalities	Skin Abnormalities	Precocious Puberty	Other Manifestations	Sanger (Blood)	Sanger (Tissue)	NGS (Blood)	NGS (Tissue)	Mutation
GS0113	M	12	ND	Yes	ND	ND	Negative	Negative (skin)	-	Negative (skin)	-
GS1029	M	17	Yes	No	No	ND	Negative	Positive (bone)	Negative	Positive (34%) C:225× T:114× (bone)	p.R201C
GS1030	F	8	Yes	Yes	ND	ND	Negative	Negative (bone)	Negative	Negative (bone)	-
GS1043	F	14	Yes (polyostotic)	Yes	ND	ND	Negative	Negative (skin)	Negative	Negative (skin)	-
GS1045	F	ND	ND	ND	ND	ND	Negative	Negative (buccal swab)	Negative	-	-
GS1048	F	8	No	No	Yes	Ovarian cyst	Negative	Negative (ovarian cyst *)	Negative	Sequence of bad quality	-
GS1050	F	ND	ND	Yes	ND	ND	Negative	-	Negative	-	-
GS1066	F	ND	ND	ND	ND	ND	Negative	-	-	-	-
GS1072	F	10	ND	Yes	Yes	Ovarian cyst	Negative	Negative (ovarian cyst)	Negative	Negative (ovarian cyst)	-
GS1077	F	ND	ND	Yes	ND	ND	Negative	Negative (buccal swab)	Negative	-	-
GS1079	M	ND	ND	ND	ND	ND	Negative	-	Negative	-	-
GS1081	M	29	Yes	Yes	Yes	Hepatic affection	Sample of bad quality	Negative (liver)	Sample of bad quality	Negative (liver)	-
GS1082	F	ND	ND	ND	ND	ND	Negative	-	Negative	-	-
GS1087	F	ND	ND	Yes	ND	ND	Negative	Negative (skin)	Negative	Negative (skin)	-
GS1091	F	ND	ND	ND	ND	ND	Negative	-	Negative	-	-
GS1098	F	ND	ND	ND	Yes	ND	Negative	-	Sequence of bad quality	-	-
GS1102	M	0	Yes	Yes	ND	Hyperthyroidism	Negative	Negative (skin)	Positive (5%) G:345× A:18×	Positive (4%) G:388× A:15× (skin)	p.R201H
GS1105	M	8	Yes (polyostotic)	ND	ND	ND	Negative	Negative (bone *)	Positive (3%) G:230× A:8×	Negative (bone *)	p.R201H
GS1109	F	1	Yes	Yes	ND	Hyperthyroidism	Negative	Positive (thyroid)	Positive (1%) G:256× A:2×	Positive (25%) G:207× A:71× (thyroid)	p.R201H
GS1128	F	29	Yes	No	No	ND	-	Positive (bone)	-	-	p.R201C
GS1144	F	9	ND	Yes	Yes	ND	Negative	-	Negative	-	-
GS1172	M	9	Yes (polyostotic)	Yes	No	ND	Negative	-	Negative	-	-
GS1176	F	3	ND	Yes	No	ND	Negative	-	-	-	-
GS1178	F	11	Yes	Yes	Yes	ND	Negative	-	-	-	-
GS1189	M	21	Yes	No	Yes	Hydrocephaly	Negative	Positive pituitary *; Negative bone *	Negative	Sequence of bad quality (G:13× A:14×) *	p.R201H
GS1224	M	29	Yes	Yes	No	Nephrotic syndrome	-	Sample of bad quality (kidney *)	-	Sequence of bad quality (kidney *)	-
GS1228	F	3	Yes (polyostotic)	Yes	ND	ND	Negative	Negative (skin)	Negative	Positive (1%) C:215× T:2× (skin)	p.R201C
GS1240	F	ND	ND	ND	ND	ND	-	Negative	-	Negative	-
GS1241	F	1	Yes	ND	ND	ND	-	Negative (bone)	-	Negative (bone)	-
GS1255	F	10	ND	Yes	Yes	Ovarian cyst	Negative	-	Negative	-	-
GS1258	F	0	ND	ND	ND	Neonatal Cushing’s syndrome	Negative	Negative (skin)	Negative	Negative (skin)	-
GS1281	M	ND	ND	ND	ND	ND	-	Negative (skin)	-	Positive (1%) C:177× T:2× (skin)	p.R201C
GS1297	M	14	Yes (polyostotic)	No	No	GH and prolactin hypersecretion	-	Negative (bone)	-	Positive (27%) C:208× T:78× (bone)	p.R201C
DX1180	F	6	No	Yes	Yes	ND	Negative	-	Negative	-	-
GS1304	F	9	Yes (monostotic)	No	Yes	ND	Negative	-	Negative	-	-
GS1311	F	9	Yes	Yes	Yes	Ovarian cyst	Negative	-	Negative	-	-
GS1312	F	9	ND	Yes	Yes	ND	Negative	-	Positive (1%) G:225× A:4×	-	p.R201H
GS1313	F	ND	ND	ND	ND	ND	-	Negative	-	Negative	-
GS1329	F	5	ND	ND	Yes	ND	Negative	-	Negative	-	-
GS1333	M	68	Yes (polyostotic)	ND	ND	ND	Negative	-	Negative	-	-

## Data Availability

The data presented in the study are deposited in the European Nucleotide Archive (ENA) repository (https:/www.ebi.ac.uk/ena, accessed on 18 December 2023), accession number PRJEB71103.

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
