# Peer review of "Choosing the Best Tissue and Technique to Detect Mosaicism in Fibrous Dysplasia/McCune–Albright Syndrome (FD/MAS)"

_genes, 2024, doi:10.3390/genes15010120_

Round 1
Reviewer 1 Report
Comments and Suggestions for Authors
General Comments:
The paper deals with genetic mosaicism and presents a method for its detection using next-generation sequencing (NGS), with a comparison with the conventional technique of Sanger sequencing. Good results have been obtained. The topic is important is genetic engineering, and the presented study is useful. However, the presentation is in need for details.
Specific Comments:
1. The terms “frequency” and “low frequency” should be clearly defined and referenced.
2. It is unclear why NGS performs better at low frequency. Explanation and details are needed.
3. There should be a clear analysis of error. Is the error rate dependent on the dataset size?
Author Response
- The terms “frequency” and “low frequency” should be clearly defined and referenced.
When we mention the term “low frequency”, we are referring to that frequency below the detection limit of Sanger sequencing. This is has been detailed and better explained in line 163-164.
- It is unclear why NGS performs better at low frequency. Explanation and details are needed.
There may have been a misunderstanding. We meant to say that NGS detects mutations well at both low and high frequencies, but that compared to Sanger and other techniques, NGS is more sensitive and capable of detecting mutations at lower frequencies. If this is not clear in the text, could you specify the lines so we can adapt the text? Thank you.
- There should be a clear analysis of error. Is the error rate dependent on the dataset size?
We are not sure to what error rate are you referring to.
When we talk about the possibility of errors when using NGS, we refer to variant calling errors, i.e., the possibility of not detecting (calling) a variant at the VCF (variant call format) file or detecting/calling a variant that it is not real, but a technical error. And these two options are related to reading depth. This is discussed in lines 211-223 of the manuscript.
Reviewer 2 Report
Comments and Suggestions for Authors
Authors reported interesting and well written study “Mosaicism in fibrous dysplasia/McCune-Albright Syndrome (FD/MAS): choosing the best tissue and technique”. However, there are a few points which must be clarified in your manuscript:
- It would be interesting for readers to further expand the Introduction on the clinical picture of patients with MAS (https://pubmed.ncbi.nlm.nih.gov/36011254/)
- Additionally, authors should expanded Discussion why their research is different from other previous studies. Authors should clearly address this issue, which allows to better understand the actual added, new information of their study, compared to knowledge already gathered by previous studies.
- Finally, I advise you to further expand the author's recommendations and the conclusions in which direction future clinical studies should be directed.
Author Response
- It would be interesting for readers to further expand the Introduction on the clinical picture of patients with MAS (https://pubmed.ncbi.nlm.nih.gov/36011254/)
As reviewer 3 asked for a shorter introduction and you suggested to introduce more clinical data we indicate and reference the current guidelines for this disease in line 79 (PMID: 31196103) where most important clinical features of FD/MAS are gathered.
- Additionally, authors should expanded Discussion why their research is different from other previous studies. Authors should clearly address this issue, which allows to better understand the actual added, new information of their study, compared to knowledge already gathered by previous studies.
Nowadays there are several techniques that can be used to study GNAS in patients with FD/MAS suspicion (PMID: 31196103, 28334704, 15096559, 36593655).
If we review the published works (PMID: 28334704, 15126527, 15096559), some speak of sensitivity in terms of the ability to detect the alteration in patients with suspected FD/MAS (clinical sensibility), while others speak of sensitivity as the ability to detect the variant in serial dilutions (technical sensibility). In some articles they work with blood samples, others with fresh biopsy, with paraffin-embedded tissue and, recently, with ccfDNA (PMID: 31196103).
We consider that in order to be able to compare some techniques with others or the suitability of some tissues over others, it is important to perform it on the same samples (or samples from the same person), as can be the work we have developed.
It is not out of the question that, among the future projects that could be considered, would be the possibility of an EQA (external quality assessment) comparing samples and laboratories. In this way, the technical capacity of the reference laboratories could also be improved.
- Finally, I advise you to further expand the author's recommendations and the conclusions in which direction future clinical studies should be directed.
We have already included that current researches are working to further improve the diagnosis of the syndrome. Regarding future directions, it would be necessary to update and improve current guidelines and to continue searching for new techniques that could improve the genetic results obtained. This proposal has been included in lines 263-267.
Reviewer 3 Report
Comments and Suggestions for Authors
Please see the attached review.

Moderate editing of English language required
Author Response
The Authors us a term fibrous dysplasia/McCune-Albright Syndrome (FD/MAS; OMIM#174800). Although it has been used, the disease is rather McCune-Albright Syndrome as the cited OMIM reference.
Since 2017 (PMID: 28243882) the international consortium for the study of these diseases (https://www.icfdmas.com/) promotes disease to be is referred as FD/MAS. Although OMIM#174800 refers to McCune-Albright syndrome as the main term, it does include FD among the other entities represented in the entry. Besides, the current guidelines for the management of FD/MAS (PMID: 31196103) do name the syndrome this way and reference to the same OMIM.
Abbreviations should be explained when first mentioned in the text.
Abbreviations have been reviewed and explained when necessary (highlighted in red in the text).
Introduction is too long, part of it rather belongs to Discussion.
Regarding your comment on the introduction, as reviewer 2 asked to expand it and you suggest shortening it, we have decided to maintain as it is. If you prefer to move some of the information to the discussion, please tell us which one are you thinking on.
Round 2
Reviewer 1 Report
Comments and Suggestions for Authors
The Authors have carefully addressed the Reviewer’s comments. Clarifications have been provided.
The current version of this manuscript is useful and suitable for publication.
Author Response
Thank you very much for appreciating our work
Reviewer 3 Report
Comments and Suggestions for Authors
Please see the attached review.

Minor editing of English language required
Author Response
Thank you very much for your comments.
An adaptation of the paragraph has been included in the text (highlighted in red) and English has been reviewed by a native speaker.